# INVESTIGATING CNNS' LEARNING REPRESENTATION UNDER LABEL NOISE

## ABSTRACT

Deep convolutional neural networks (CNNs) are known to be robust against label noise on extensive datasets. However, at the same time, CNNs are capable of memorizing all labels even if they are random, which means they can memorize corrupted labels. *Are CNNs robust or fragile to label noise?* Much of researches focusing on such memorization uses class-independent label noise to simulate label corruption, but this setting is simple and unrealistic. In this paper, we investigate the behavior of CNNs under class-dependently simulated label noise, which is generated based on the conceptual distance between classes of a large dataset (i.e., ImageNet-1k). Contrary to previous knowledge, we reveal CNNs are more robust to such class-dependent label noise than class-independent label noise. We also demonstrate the networks under class-dependent noise situations learn similar representation to the no noise situation, compared to class-independent noise situations.

## 1 INTRODUCTION

Deep convolutional neural networks (CNNs) excel in supervised image classification tasks (Krizhevsky et al. (2012)). Representation learned from such tasks can be transfer to other tasks, including object detection (Ren et al. (2015); Liu et al. (2016); Redmon et al. (2016)) and semantic segmentation (Chen et al. (2015); Badrinarayanan et al.). Furthermore, if the training dataset is sufficiently larger, CNNs can improve the performance in classification, or learn better transferable representation, even if some labels are corrupted (Li et al. (2017); Sun et al. (2017); Mahajan et al. (2018)).

However, recent CNNs have far more parameters than their training samples. Therefore, the networks can memorize all the training data even if all labels are randomly replaced with the wrong ones (Zhang et al. (2017); Arpit et al. (2017)). This capability may degrade CNNs' performance under the label-corrupted situation, thus learning methods against label noise have been studied.

Are CNNs robust or fragile to label noise? To investigate this question, we need to adopt noisy labels in controlled experiments. In previous work, both natural and synthetic noise have been used to research label corrupted situations. Natural noise appears in generally every dataset, and it comes from, for instance, annotators' mislabeling (Deng et al. (2009)) or their varieties (Dgani et al. (2018)). Some researchers have been proposed robust training methods under this type of noise (Lee et al. (2017); Jiang et al. (2018); Tanaka et al. (2018)). However, natural noise is uncontrollable, in other words, the relationship between the magnitude of noise and CNNs' performance has been unknown.

On the other hand, synthetic noise simulates natural one by stochastically replacing ground truth labels with others. Class-independent uniform label permutation is a common setting (Jiang et al. (2018); Han et al. (2018b)), yet some researchers use class-dependent label permutation, which is considered as more realistic situation (Natarajan et al. (2013); Ghosh et al. (2017); Patrini et al. (2017); Han et al. (2018a)). Previous research has mainly adopted MNIST (10 classes, 60,000 training samples, LeCun et al. (1998)) or CIFAR-10/100 (10 and 100 classes, 50,000 training samples, Krizhevsky et al. (2012)), and these datasets lack pre-defined conceptual relationships between classes. This limitation results in simplified noise simulation on such datasets, although synthetic noise enables researchers to research the relationship between the noise magnitude and the performance of networks.

To investigate whether CNNs are robust or fragile to label corruption, we propose to use simulated noise considering possible mislabeling on ImageNet-1k (Russakovsky et al. (2015)) to complement the disadvantages. Exploiting ImageNet-1k's conceptual hierarchy, we can divide its 1,000 labels into some clusters. We use these clusters to generate class-conditional label noise. We train several networks on the training dataset with and without corrupted labels. Then we evaluate the performance of the networks on the original validation set, the robustness of the networks against adversarial perturbation (Szegedy et al. (2013); Poursaeed et al. (2018)), and their learned representation using transfer learning, canonical correlation analysis (Raghu et al. (2017); Morcos et al. (2018b)).

In this paper, we show the performance of CNNs trained on such synthesized noise considering possible mislabeling is better than uniformly synthesized noise, which is contrary to previous research (Ghosh et al. (2017); Patrini et al. (2017)). Besides, models trained on class-dependent label noise are more robust to adversarial perturbation than ones trained on class-independent label noise. We also demonstrate CNNs trained under class-conditionally noisy conditions learn similar features to ones trained under the clean condition. As a result, even when 80% of labels are class-dependently corrupted, CNNs can learn useful representation for transfer learning. Meanwhile, we demonstrate class-independent noise leads models to learn different representation from ones trained with data with clean labels or label noise considering conceptual hierarchy. These differences can be attributed to the property of categorical cross entropy loss, which is a well-used loss function for image recognition tasks. We believe using class-independent noise is not a suitable protocol to investigate the CNNs' tolerance in practical situations.

## 2  LABEL NOISE SIMULATION

In this section, we describe some methods to synthesize label noise. To simulate label noise, a noise transition matrix $N$ is used. Specifically, when the ground truth label of a sample is $i$, it is replaced with $j$ with probability $N_{ij}$, where $\sum_j N_{ij} = 1$.

### 2.1  CLASS-INDEPENDENT NOISE

The simplest noise generation method is class-independent and uniform noise (Ghosh et al. (2017); Patrini et al. (2017); Jiang et al. (2018)). To generate this type of noise, we replace the label of each image with one of the other with a probability $p$. We refer to this probability as *noise fraction* (Jiang et al. (2018)). In this case, the transition matrix is

$$N_{ij} = \begin{cases} 1 - p & \text{if } i = j \\ \dfrac{p}{\dim \boldsymbol{N} - 1} & \text{otherwise.} \end{cases} \tag{1}$$

### 2.2  CLASS-DEPENDENT NOISE

Natural noise is class dependent. To simulate this nature, Patrini et al. (2017) and Han et al. (2018a) stochastically swaps similar-class labels, e.g. CAT → DOG not CAT → TRUCK, using dual-, tri- or block-diagonal transition matrices. Transition matrices in the previous research were manually constructed.

We use the inter-class distance (Deng et al. (2010)) as a measure of class dissimilarity to generate a transition matrix. Each class in ImageNet is a leaf (synset) of a conceptual tree of WordNet (Miller (1995)). The inter-class distance between class $i$ and $j$ is the number of hops to reach the closest shared ancestor $k$ between them (Figure 1). If the numbers of hops from $i$ to $k$ and from $j$ to $k$ differ, we use the larger one to make the distance symmetric (Figure 2 **(a)**).

To generate label noise, we clustered 1,000 categories of ImageNet-1k labels into subgroups by Ward's hierarchical clustering. In this case, the transition matrix is block diagonal. $k$th diagonal block $\boldsymbol{N}^{(k)}$ of the transition matrix $\boldsymbol{N}$ is written as

$$N_{ij}^{(k)} = \begin{cases} 1 - p & \text{if } i = j \\ \dfrac{p}{\dim \boldsymbol{N}^{(k)} - 1} & \text{otherwise.} \end{cases} \tag{2}$$

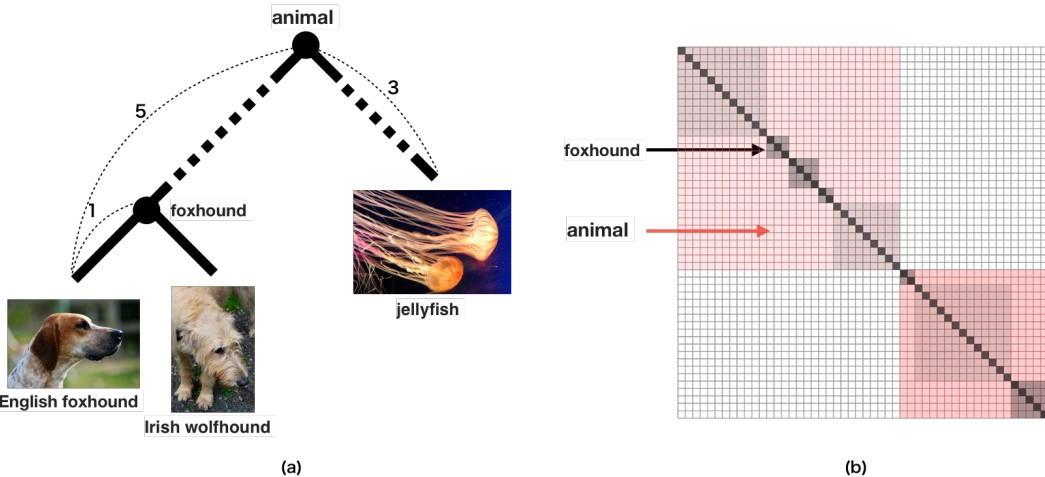

(a)
(b)

Figure 1: **(a)** An intuitive illustration of the inter-class distance. The distance between "English foxhound" and "Irish wolfhound" is 1 and "English foxhound" and "jellyfish" is 5. **(b)** An intuitive illustration of the noise transition matrix $N$ used in this paper. Darker color corresponds to higher probability. When the label of a sample is $i$, we replace its label with $j$ with probability $N_{ij}$. The number of cluster can be changed (gray corresponds to 8 and red corresponds to 2).

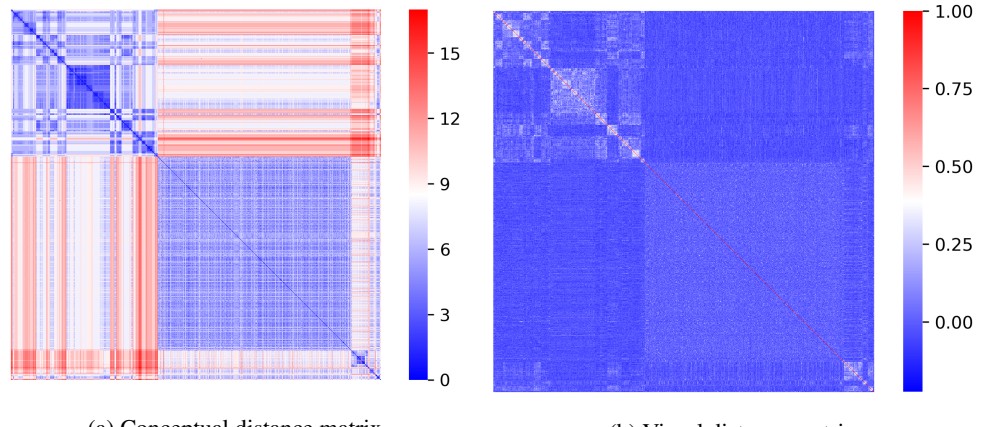

(a) Conceptual distance matrix
(b) Visual distance matrix

Figure 2: **(a)** Conceptual distance matrix between all classes in ImageNet-1k. We use this distance to cluster the classes into some subgroups for noise generation. **(b)** Cosine distance matrix between vectors in the final fully connected layer. This matrix shows a similar structure to the inter-class distance matrix. This resemblance means conceptual similarity correlates with visual similarity. Based on this observation, we use conceptual distance matrix to generate label noise.

In the rest of this paper, we refer label noise generated by this noise transition matrix as *class-dependent noise*.

Indeed, the conceptual similarity resembles to the visual similarity (Deselaers & Ferrari (2011)). In Figure 2 **(b)**, we show an inter-class cosine distance matrix $D$, where $D_{ij} = \cos \text{distance}(W_{i,:} W_{j,:})$ and $W$ is the weight of that layer of ResNet-50. Here, $W_{i,:}$ is a $2048\,\text{dim}$ vector of the class $i$. The conceptual distance matrix Figure 2 **(a)** and the cosine distance matrix Figure 2 **(b)** correlate, therefore, we can assume this noise generation method simulates possible mistakes in label annotation: annotators may often mislabel "English foxhound" as "Irish wolfhound" but rarely (or never) "English foxhound" as "jellyfish" (Figure 1).

## 3 EXPERIMENTS

To investigate whether CNNs are noise tolerant or noise sensitive, we first train CNNs with no label noise, class-dependent label noise and class-independent label noise settings (Section 3.2). We also generate adversarial perturbations in each setting (Section 3.3). We further investigate their internal representation using several methods (Section 3.4).

### 3.1 COMMON EXPERIMENTAL SETUP

We employ ResNet-50 (He et al. (2016)) as a standard network following Morcos et al. (2018a). We also use VGG-19 (Simonyan & Zisserman (2014)) and DenseNet-121 (Huang et al. (2016))[1].

To train these networks, we use SGD with a momentum of 0.9 and weight decay of $1.0 \times 10^{-4}$. Without specification, we fix the mini-batch size to 512 and adjust the initial learning rate in proportion to the mini-batch size from the original learning rate (Jastrzębski et al. (2017); Smith & Le (2017)). For example, the initial learning rate of ResNet-50 in He et al. (2016) is 0.1, and the mini-batch size is 256, therefore, in our experiment, the learning rate is set to $0.2 \ (= 0.1/256 \times 512)$. We train networks for 90 epochs and divide the learning rate by 10 at the 30th and 60th epoch. We use categorical cross entropy as the loss function.

#### 3.1.1 IMAGENET-1K

ImageNet-1k is a 1000-class subset of ImageNet for ILSVRC image classification task (Russakovsky et al. (2015)). We use the dataset for ILSVRC 2012. ImageNet-1k consists of 1.2 million images for training and 50,000 images for validation. We use this validation set for testing.

In the experiments, all the input images are standarized. In the training phase, before the standarization, we randomly crop the input images into 224px $\times$ 224px[2] and randomly apply horizontal flip to augment the inputs. In the evaluation phase, we resize images into 256px $\times$ 256px and then center-crop them into 224px $\times$ 224px.

### 3.2 TRAINING CNNS WITH NOISY LABELS

To examine how label corruption influences the performance of networks, we trained several networks on noisy ImageNet-1k and evaluated on a clean evaluation dataset.

According to Patrini et al. (2017) and Han et al. (2018b), learning with class-dependent noise is more difficult (e.g., lower test accuracy) than learning with class-independent noise on CIFAR-10/100. We conducted ImageNet-1k counterpart. For class-dependent noise, we set the number of clusters to 50.

Figure 3 shows test accuracy curves of ResNet-50 trained under no noise, class-dependent noise when the number of clusters is 50 and class-independent noise situations. In the case of ImageNet-1k, test accuracy at the last epoch on 40% class-independent noise is 4 % worse than that of 40% class-independent noise. Furthermore, when the noise fraction is 80%, this gap reached 28%. This outcome indicates CNNs are robust to class-dependent noise, compared to class-independent noise. This property can be seen in not only ResNet-50 but VGG-19 and DenseNet-121 (Figure 4). Also, this property is the case when we change the number of clusters from 50 to 10 (Figure 7 in Appendix).

### 3.3 ADVERSARIAL PERTURBATION

Adversarial perturbation $\varepsilon$ is a small perturbation such that a classifier network classifies an input $x$ and $x + \varepsilon$ into different classes (Szegedy et al. (2013)). We use Generative Adversarial Perturbation (Poursaeed et al. (2018)), which uses a generative network to create such perturbation $\varepsilon$. Besides, we adopt a non-targeted universal perturbation setting, where the generator learns to output a single perturbation $\varepsilon$ so that the classifier degrades the performance as worse as possible.

---

[1] We used `torchvision.models`' implementations.
[2] Precisely, we apply `torchvision`'s `transforms.RandomResizedCrop`.

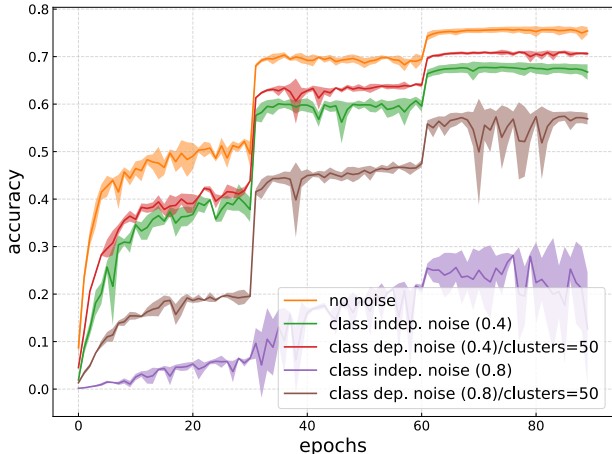

Figure 3: **Class-dependent noise results in less performance decrease than class-indepednet noise.** Test accuracy with ResNet-50 on ImageNet-1k validation set with class-independent and class-dependent noise. `clusters` is the number of clusters for label noise generation described in section 2.2. We report mean and standard deviation of test accuracy along three runs.

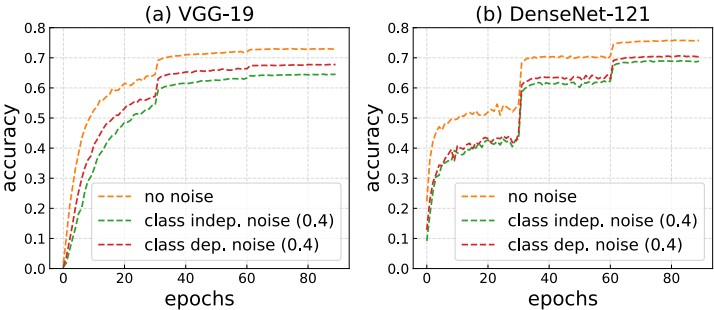

Figure 4: Test accuracy of VGG-19 and DenseNet-121 trained with class-dependent noise, with class-independent noise and without noise. Class-dependent noise results in less performance decrease not only on ResNet-50 but on other networks.

To generate a perturbation, we trained the generator on 3200 images in ImageNet-1k training set for 10 epochs and evaluated on the whole validation set. As the implementation of Poursaeed et al. (2018), the generator is optimized by Adam (Kingma & Ba (2015)) with the learning rate of 0.0002.

In Table 1, we report fooling rate, the ratio of inconsistency between the network's predictions corresponding to $x$ and $x + \varepsilon$. These results show models trained on class-independent noise are more fragile to adversarial perturbation than models trained on no noise or class-dependent noise. In particular, when the noise fraction is $80\%$, class-dependent noise results in marginal fooling rate arise from no-noise, while class-independent yields drastic arise.

### 3.4 LEARNED REPRESENTATION

Why models trained on class-dependent noise show different behavior from models trained on no noise or class-dependent noise? Here, we investigate the learned features using the performance of canonical correlation analysis, transfer learning and adversarial perturbation.

### 3.4.1 CANONICAL CORRELATION ANALYSIS

We use canonical correlation analysis to compare the representation of a pair of networks (Raghu et al. (2017); Morcos et al. (2018b)). Specifically, this method compares a pair of output features of two layers using CCA, which is invariant to affine transformation. The inputs corresponding to the

Table 1: **Models trained on class-dependent noise is more robust to adversarial perturbation than models trained on class-independent noise.** Fooling rate (%) for adversarial perturbation (ResNet-50). We used three pre-trained models for each experiment and report `mean±(std)`.

| MODEL | FOOLING RATE (%) |
|---|---|
| clean model | $74.6 \pm 4.5$ |
| class-dep model (40 %) | $73.9 \pm 4.2$ |
| class-dep model (80 %) | $75.2 \pm 2.9$ |
| class-indep model (40%) | $88.0 \pm 1.9$ |
| class-indep model (80%) | $92.3 \pm 4.1$ |

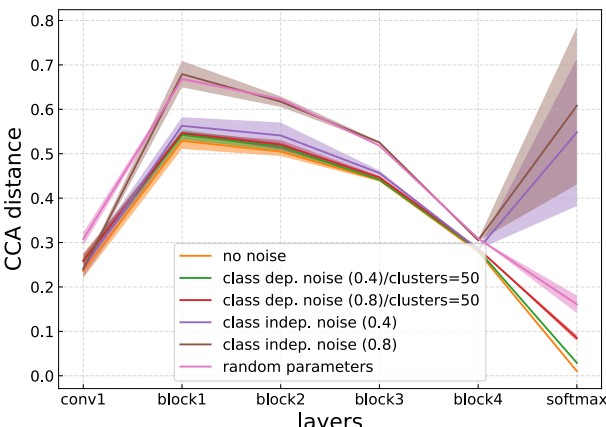

Figure 5: **CNNs trained on class-dependent label noise learn similar representation of CNNs trained on clean data.** Projection-weighted CCA distance between ResNet-50s and a cleanly trained ResNet-50 at each component. These results indicate class-independent noise leads CNNs to learn unnatural representation in the final layer. We report mean and std of distance along three different models.

features are a set of images. We adopted projection-weighted CCA (Morcos et al. (2018b))[3]. The distance between two output feature matrices $\boldsymbol{L}^{(1)}$ and $\boldsymbol{L}^{(2)}$ are defined as

$$d(\boldsymbol{L}^{(1)}, \boldsymbol{L}^{(2)}) = 1 - \sum_{i=1}^{c} \alpha_i \rho^{(i)}, \qquad (3)$$

where $\rho^{(i)}$ is the $i$th canonical correlation coefficient and the weights $\alpha_i$ are normalized values of $\tilde{\alpha}_i = \sum_j |\boldsymbol{h}_i \cdot \boldsymbol{L}_{i,:}^{(1)}|$. $\boldsymbol{h}_i$ is the $i$th canonical correlation vector.

We sampled 3,600 images from ImageNet-1k training dataset as inputs and measured distance between ResNet-50 models trained under several settings and a model trained on the clean ImageNet-1k. We also measured the distance between randomly initialized models and the clean reference model. The results in Figure 5 suggest models trained on class-independent noise learn different representation from that of models trained on clean or class-dependent noise. This contrast is significant in the final output of the network (`softmax`). This difference indicates class-independent noise disturb the structure of the last layer shown in Figure 2 **(b)**, while class-dependent noise keeps it. Furthermore, the curve of class-independent noise shows less convergence than those of no noise, class-dependent noise. In addition, under 80% class-independent noise, the models learn similar representation to randomly initialized models at `block1`, 2, 3 and 4.

---

[3]We used an implementation of `https://github.com/moskomule/cca.pytorch`.

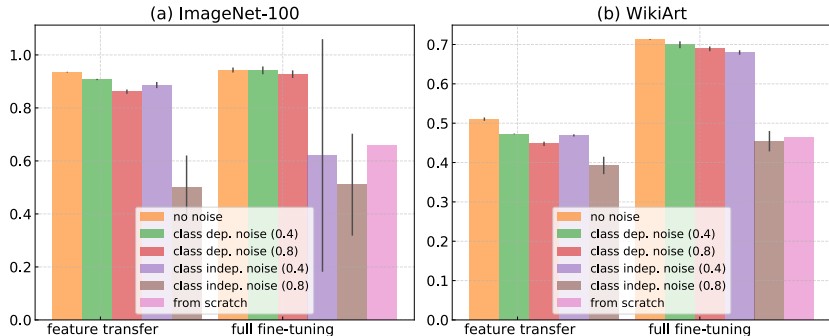

Figure 6: **Class-independent noise degrades the performance of transfer learning compared to class-dependent noise.** Test accuracy in transfer learning (ResNet-50) on ImageNet-100 and WikiArt. We pre-trained models on ImageNet-1k and replaced the final layers with random ones. We tested three pre-trained models for each experiment and report mean and standard deviation. For from scratch, we report the result of a single run in the full fine-tuning setting.

### 3.4.2 TRANSFER LEARNING

We adopt transfer learning as a tool to evaluate the learned representation. We train networks on ImageNet-1k and then replace the final fully connected layer with a random one that the output dimension is the number of classes of a target dataset. Following Mahajan et al. (2018), we use *full network fine-tuning* and *feature transfer*. In the former setting, we fine-tune the entire network, and, in the latter, we train only the final layer. We use two datasets for this task: ImageNet-100 from a similar domain to ImageNet-1k and WikiArt from a different domain.

**ImageNet-100** To investigate the transferability to a similar domain to the source, we created ImageNet-100, namely, 100-class subset from ImageNet 10k (Deng et al. (2010)). There is no overlapped category between ImageNet-1k and ImageNet-100. Each class has more than 1,000 training samples and 50 test samples. More detailed information on this dataset is described in Appendix A.

**WikiArt** As a different-domain dataset from ImageNet-1k, we adopt WikiArt dataset, which consists of 81 thousand paintings drawn from the 15th to 21st century. Following Elgammal et al. (2018), we treated this as a 20-class dataset and sampled 5.5% of paintings as test data. The details are in Appendix B.

In the experiments, we randomly selected a small amount of training data as validation data to select appropriate hyper-parameters. For faster convergence, we used Adam optimizer (Kingma & Ba (2015)) and trained networks for ten epochs on ImageNet-100 and 30 epochs on WikiArt, respectively.

The results shown in Figure 6 demonstrate the magnitude of class-dependent label noise in the source data degrades the performance on the target data. However, test accuracy of such transferred models is better than models trained from scratch and ones whose source data are class-independently corrupted, in both feature transfer and full fine-tuning settings. In the full fine-tuning setting, models pre-trained on data with 40% class-independent label noise result in worse performance than ones pre-trained on class-dependent label noise. Moreover, when the models are transferred from 80% class-independent label noise situation, the performance on both datasets are worse than when the models are trained from scratch. These results may be related to the difference of the learned representation between models trained under class-dependent and class-independent label corruption, shown in Section 3.4.1 CANONICAL CORRELATION ANALYSIS.

## 4 DISCUSSION

Why does class-dependent noise affect less than class-independent noise? We think there are two reasons: class-dependent noise is more informative, and it avoids the loss value getting too large.

When class-dependent noise swaps a ground truth label with a wrong one, it is still a similar class to the original. Thus, the network can learn "which cluster the sample belongs to". This idea is related to the soft label (Hinton et al. (2015)), though in our case, the label is "hard". Contrary to this, class-independent noise conveys no information.

The other reason results from the property of categorical cross entropy loss. When the label of sample $x$ is $i$, the loss value can be written as $-\log[f(x)]_i$, where $f(x)$ is the corresponding softmax output. Therefore, when a CNN predicts $x$ as $i$ with weaker confidence, the penalty gets larger. Since the wrong label corrupted by class-dependent noise belongs to the same cluster as the ground truth, $[f(x)]_i$ is relatively large (c.f. Figure 2 **(b)**). However, in the case of class-independent noise, the wrong label has nothing to do with the ground truth, and if the ground truth and the corrupted label are irrelevant, $[f(x)]_i$ should be small. Thus, the loss value gets larger, which leads the network to a worse solution.

Also, our finding can be applicable to the quality control of annotation of data. Our results show class-dependent noise is more favorable than class-independent noise. Inexperienced but honest annotators will yield class-dependent noise, while lazy and malicious annotators may randomly annotate the labels and will yield class-independent noise. Therefore, according to our results, the administrators of the annotation need to exclude such workers.

## 5   CONCLUSION

In this paper, we investigated the relationship between label noise, the performance and representation of CNNs in image classification tasks. We used ImageNet-1k with simulated noise which includes class-independent noise and class-dependent noise considering conceptual similarity. We examined such noise considering possible mislabeling causes less performance decrease and more robustness against adversarial perturbation compared to class-independent noise. Besides, we investigated the internal representation of CNNs trained with and without label corruption. Experiments showed networks trained on class-independently noisy data learn different representation from ones trained on clean or class-conditionally noisy data.

Some previous research on label-noise-tolerant learning methods has used class-independent noise. However, as we revealed in this research, this noise setting is so artificial and straightforward that such methods may not be effective against real noise. Meanwhile, our results suggest plain CNNs themselves can be robust against real noise. This property should be good news for practitioners. Nevertheless, it is also shown noise considering possible mislabeling still somewhat degrades the performance of networks. Thus, how to avoid the effect of label noise is still a remaining problem.

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

APPENDIX

## A  IMAGENET 100

To create ImageNet 100 dataset, we first selected synsets of ImageNet 10k which contains more than 1200 images. Then we removed overlapped classes with ImageNet 1k, descendant classes of "psychological feature" (e.g. "sumo") and non-leaf classes (e.g. "animal"). We randomly sampled 150 classes from the remainders, and manually excluded abstract classes such as "playpen, pen" and finally chose 100 classes. From each class, we separated 50 images as validation data.

## B  WIKIART

Following Elgammal et al. (2018), we used 20 categories as Table 2. Some classes were merged.

Table 2: Refined WikiArt dataset used in our experiment.

| CLASS | MERGED CLASSES |
|---|---|
| Early Renaissance | - |
| High Renaissance | - |
| Mannerism and Late Renaissance | - |
| Northern Renaissance | - |
| Baroque | - |
| Rococo | - |
| Romanticism | - |
| Impressionism | |
| Post-Impressionism | Post Impressionism & Pointillism |
| Realism | Realism & Contemporary Realism & New Realism |
| Art Nouveau | - |
| Cubism | Cubism & Analytical Cubism & Synthetic Cubism |
| Expressionism | - |
| Fauvism | - |
| Abstract-Expressionism | Abstract Expressionism & Action Painting |
| Color field painting | - |
| Minimalism | - |
| Naïve art-Primitivism | - |
| Ukiyo-e | - |
| Pop-art | - |

## C  EFFECT OF NUMBER OF CLUSTERS

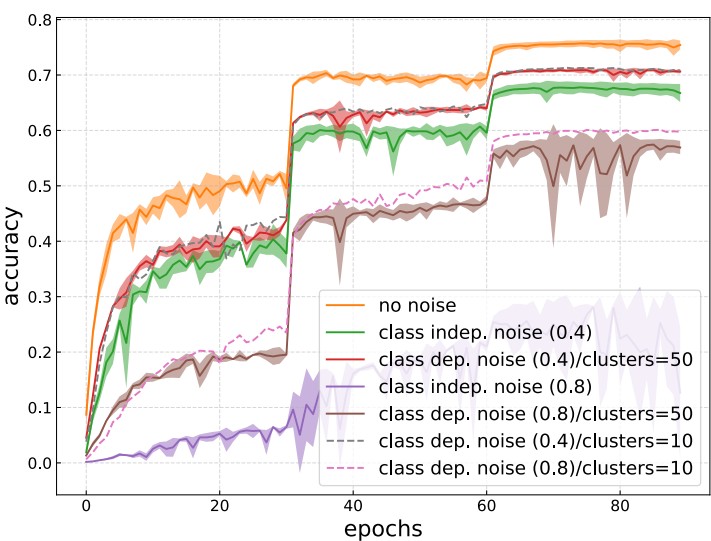

Figure 7: Test accuracy with ResNet-50 on ImageNet-1k validation set with class-independent and class-dependent noise. Extending Figure 3, we report the results when the number of clusters is 10. The results show the difference of the number of clusters is ignorable.

