# OpenReview forum: "Investigating CNNs' Learning Representation under label noise"
_ICLR.cc/2019/Conference_

### Official Review · AnonReviewer3 · 2018-11-01
**CCN model and IDN model should be the focus in learning with noisy labels.**

**Rating:** 5
**Confidence:** 5

**Review:**

This paper demonstrates that CNNs are more robust to class-relevant label noise. They argue that real-world noise should be class-relevant.

Pros:

1. The authors find a new angle to exploit robust learning with noisy labels.

2. The authors perform numerical experiments to demonstrate the effectiveness of their proposal. And their experimental result support their previous claims.

Cons:

We have two questions in the following.

1. Basic definition: in learning with noisy labels, there are two basic models. First, most research focuses on class-conditional noise (CCN) model [1]. Second, recent research explore a bit on instance-dependent noise (IDN) model [2, 3]. As far as I know, there is no class-irrelevant label noise and class-relevevant label noise. In CCN mode, people would like to use symmetric noise and asymmetric noise as a basic benchmark to conduct experiments.

2. Motivation: The authors want to claim CNNs are more robust to such realistic label noise than class-irrelevant label noise. However, they make one mistake. They do not have a clear definition about realistic label noise. In my mind, I believe Clothing1M [4] should be realistic label noise dataset.

By the way, in learning with noisy labels, there are two kinds of research. First, people propose new robust methods for CCN model. Second, people propose new robust methods for IDN models. Proposing new setting should be encouraged. However, the setting and conclusion should be reasonable.

References:

[1] D. Angluin and P. Laird. Learning from noisy examples. Machine Learning, 1988.

[2] A. Menon, B. Rooyen, and N. Natarajan. Learning from binary labels with instance-dependent corruption. Machine Learning, 2018.

[3] J. Cheng, T. Liu, K. Ramamohanarao, D. Tao. Learning with bounded instance-and label-dependent label noise. arxiv 1709.03768, 2017.

[4] T. Xiao, T. Xia, Y. Yang, C. Huang, and X. Wang. Learning from massive noisy labeled data for image classification. In CVPR, 2015.

---

> ### Author Response · Authors · 2018-11-21
> **We revised some problems on expressions**
>
> We highly appreciate your efforts for reviewing. Your comments with citations are helpful. To sum up, we modeled the label noise so that each ground truth label is replaced with the wrong one randomly sampled from the same group, which is clustered based on conceptual distance. CNNs show robustness to class-dependent noise compared to class-independent noise in the perspective of test accuracy and fooling rate of adversarial examples. Moreover, we showed the models trained on class-dependent label noise learn similar representation to the models trained under no-label noise settings, on the other hand, the ones trained on class-independent label noise learn different representation.
>
> > As far as I know, there is no class-irrelevant label noise and class-relevevant label noise. In CCN mode, people would like to use symmetric noise and asymmetric noise as a basic benchmark to conduct experiments.
>
> We used "class-irrelevant noise" only in the abstract for those who are unfamiliar with this problem. If necessary, we will correct it to "class-independent" in the revised version.
> In our research, we used ImageNet-1k, which contains 1000 classes, and constructing hand-crafted asymmetric noise is difficult. Therefore, we employed symmetric noise for both class-dependent and class-independent settings.
>
> > However, they make one mistake. They do not have a clear definition about realistic label noise.
>
> Thank you for pointing out the unclarity of our paper. We admit "realistic" is a too strong expression, so we change it to "class-dependent" in the revised version.

---

### Official Review · AnonReviewer2 · 2018-11-01
**A good start, but lack of sufficient support**

**Rating:** 4
**Confidence:** 5

**Review:**

This paper attempted to analyse the performance of CNN models when data is mislabelled in different manners, i.e. class dependent labels and class independent labels. It carried out several good experiments as a good start, but several points are not comprehensively studied and analysed.
1. Try to provide more direct and solid proofs on the relationship between conceptual and visual distances between class dependent labels.
2. In table 1, model trained with noise on class dependent label has lower fooling rate than model trained with clean data. Is it worth exploiting in a deeper manner?
3. In figure 5, why the curve appears so different after block 4 only? Would visualising feature maps from different layers help understand this observation?
4. In figure 6, it seems that the difference between those experiments is marginal, which contribute little to the argument of this paper.
5. About the discussion on 'cluster', it would be better if sufficient experiments and analysis can be provided.

---

> ### Author Response · Authors · 2018-11-21
> **Additional experiments support our ideas**
>
> We highly appreciate your efforts for reviewing. To sum up, we modeled the label noise so that each ground truth label is replaced with the wrong one randomly sampled from the same group, which is clustered based on conceptual distance. CNNs show robustness to class-dependent noise compared to class-independent noise in the perspective of test accuracy and fooling rate of adversarial examples. Moreover, we showed the models trained on class-dependent label noise learn similar representation to the models trained under no-label noise settings, on the other hand, the ones trained on class-independent label noise learn different representation.
>
> > Try to provide more direct and solid proofs on the relationship between conceptual and visual distances between class dependent labels.
>
> For example, measuring CCA distance between conceptual and visual distance matrices is an answer to your question?
>
> > In table 1, model trained with noise on class dependent label has lower fooling rate than model trained with clean data. Is it worth exploiting in a deeper manner?
>
> In the revised version, we add the fooling rates of models trained on 80% class-dependent and class-independent label noise. They show the difference between the fooling rate of models trained on clean labels and on 40% class-dependent label noise is marginal.
>
> > In figure 5, why the curve appears so different after block 4 only? Would visualising feature maps from different layers help understand this observation?
>
> In Figure 5, we show the CCA distance between a clean model and other models, and models trained on class-independent noise show a significant difference from the other ones. Intuitively, without label noise, each softmax output is close to a onehot vector. With class-dependent label noise,  it is also close to a onehot vector, but the elements correspond to the cluster which the predicted class belongs are non-zero, to minimize the loss function. On the other hand, with class-independent label noise, all elements of the softmax output vector needs to be non-zero thus smoothed onehot vector. We think the difference of CCA distance derives from this difference. We consider adopting visualization of feature maps for a better understanding of the observation.
>
> > In figure 6, it seems that the difference between those experiments is marginal, which contribute little to the argument of this paper.
>
> In the revised version, we add the results of 80% class-independent noise, where the difference of performance between the models trained on 80% class-dependent and 80% class-independent noise is clear.
>
>
> > About the discussion on 'cluster', it would be better if sufficient experiments and analysis can be provided.
>
> In our research, we cluster similar classes, and in the pre-revision, we only showed the test accuracy with models trained on class-dependent label noise whose number of clusters is 50. In the revised version, we add the results of models trained on class-dependent label noise whose number of clusters is 10.

---

### Official Review · AnonReviewer1 · 2018-11-02
**Interesting analysis, but not surprising results**

**Rating:** 5
**Confidence:** 4

**Review:**

The authors challenge the CNNs robustness to label noise, but when the label noise is class dependent, more realistic scenario than class independent noise.
To analyse the CNNs behavior in such a scenario, they consider the ImageNet 1k dataset, and change some labels to labels that are close according to the ImageNet 1k tree of WordNet.
The authors conduct multiple experiment to compare the effect of class dependent and class independent noise on:
* the model accuracy
* the robustness to adversarial perturbation
* the learned representation

The paper is generally well written and well structured. The analysis is sound and addresses interesting points, giving insightful results. Nevertheless, the overall conclusion is not very surprising. This work  confirms the commonly admitted fact that CNNs learn features that are visually meaningful.   Moreover, there is no significant novelty in the paper. The paper only analyses the CNNs behavior, without suggesting any new algorithm based on the observations. One specific point that seems under-investigated in my sense is the observation about the robustness to adversarial perturbations. The model with the class dependent noisy labels is in average less sensitive to the perturbations, even if this is not significant for the tested noise level. Did the authors test with different noise levels? This calls for a further analysis. It has the potential to give more insights, and probably inspire new methods to improve training robustness.

---

> ### Author Response · Authors · 2018-11-21
> **Additional experiments support our ideas**
>
> We highly appreciate your efforts for reviewing. To sum up, we modeled the label noise so that each ground truth label is replaced with the wrong one randomly sampled from the same group, which is clustered based on conceptual distance. CNNs show robustness to class-dependent noise compared to class-independent noise in the perspective of test accuracy and fooling rate of adversarial examples. Moreover, we showed the models trained on class-dependent label noise learn similar representation to the models trained under no-label noise settings, on the other hand, the ones trained on class-independent label noise learn different representation.
>
> > Did the authors test with different noise levels?
>
> Yes. In the revised version, we add the results of 80% class independent noise to Table 1, Figure 5, 6, which was missed in the pre-revised one. The results support our claims. Compared to 80% class-dependent noise, 80% class-independent noise degrades the robustness against adversarial examples (Table 1, 75.2->92.3), change the internal representation in the viewpoint of CCA distance (Figure 5) and degrade the performance of transfer learning (Figure 6, e.g. 68.5%->45.4% in the feature transfer setting on WikiArt).

---

### Meta-Review · Area_Chair1 · 2018-12-12
**Limited novelty**

**Confidence:** 4
**Recommendation:** Reject

**Metareview:**

The paper analyzes the performance of CNN models when data is mislabelled in different manners.

The reviewers and AC note the critical limitation of novelty of this paper to meet the high standard of ICLR.

AC thinks the proposed method has potential and is interesting, but decided that the authors need more works to publish.